# OpenReview forum: "From Weak Cues to Real Identities: Evaluating Inference-Driven De-Anonymization in LLM Agents"
_ICML.cc/2026/Conference — ICML 2026 regular_

### Official Review · Reviewer_YSKc · 2026-03-11

**Soundness:** 3
**Presentation:** 2
**Significance:** 3
**Originality:** 2
**Overall Recommendation:** 2
**Confidence:** 4

**Summary:**

This paper explores the capabilities of LLM based agents to perform inference-driven linkage risk, which is where identities are reconstructed by aggregating fragmented, individually non-identifying signals. Authors explore three cases: (i) the classical Netflix and AOL linkage, where they compare agents to traditional approaches, (ii) a controlled benchmark, where they show that agents perform identity leakage even for benign prompts, and (iii) real-world settings, namely on the Anthropic Interview dataset and anonymized ChatGPT logs.

The paper shows that language models either equal or exceed classical linkage risk attacks with minimal prompting and domain knowledge. It also shows that models do these types of attacks even under benign prompts, both for synthetic benchmarks and in real world settings. This can be mitigated with a privacy-aware system prompt, but this comes with utility loss.

**Compliance With Llm Reviewing Policy:**

Affirmed.

**Key Questions For Authors:**

Questions:

- For the synthetic benchmark, authors explicitly state that $D_\text{anon}$ is anonymized data and $D_\text{aux}$ is not anonymized, and that these datasets share exactly one individual. Does this "one individual overlap" remain true in the Netflix use case? Or is this singular overlap only for the benchmark dataset?
- Related to the above, for a given seed in the synthetic benchmark, e.g. INTRINSIC EXPLICIT-MK, is one pair of datasets $(D_\text{anon}, D_\text{aux})$ produced or are many dataset pairs $\\{(D_\text{anon,1}, D_\text{aux,1}), (D_\text{anon,2}, D_\text{aux,2}), \ldots, (D_\text{anon,N}, D_\text{aux,N})\\}$ produced? If many are generated, how many? What is the value of $N$? I ask because for the Netflix dataset, it seems that we have an independent $D_\text{aux}$ for every element in $D_\text{anon}$, which implies $N = 1000$.
- In Table 1, what is $T$? It seems like this was never defined.
- In Table 1, authors report the mean across 3 independent runs. Could authors also provide some metric of uncertainty as well? I ask because the classical and agent LSR are often very close, and I am wondering if this result is statistically significant.
- Table 1 shows a major difference between the GPT5 agent and the Claude 4.5 agent, especially at $m = 2, 4$. Table 2 shows a similar difference between the two models especially in the IMPLICIT intent. It would be great if authors pointed this out and explained why this happens.
- In the benchmark, it is not clear to me what the utility task is. Authors give as an example ```uploading a report to OwnCloud```, but it is not clear to me what the report is about, or if authors checked if the report is meaningful in any way.
- For the AOL setting, the agent identified 10 distinct individuals from the AOL dataset. Authors claim ```prior analyzes of the AOL release are associated with a single manually identified case```. Could authors provide a source for this claim?
- For the Scientist subset of the Anthropic Interview dataset, the agent linked 6 interviews to unique real-world research profiles. The previous work of (Li 2026) also linked 6 interviews to specific scientific works, though not explicitly to real-world research profiles. I find it odd that authors did not mention this. What do authors contribute that goes beyond what (Li 2026) did? Do they think that the 6 interviews they identified are the same as the ones in (Li 2026)?
- I find the analysis of the anonymized ChatGPT conversation logs lacking. Authors narrowed down 1,916 sessions to 30 high-risk conversations for analysis. After removing PII (how did authors do this?), authors claim that often times the agent was able to narrow down the list from 37,000 potential candidates to one candidate. It would be great if authors mentioned in how many of the 30 conversations they were able to do this, and in how many they were unable to. Along the same lines, it would be great if authors could add quantitative information to the qualitative Table 3.

Overall, I will recommend rejection, due to a lack of detail in every use case explored in the paper, the inconsistent notation that is introduced and dropped in various sections, and the author's choice to not mention that their Anthropic Interview findings are identical to (Li 2026).  I am willing to adjust my score if authors fix these issues and/or correct any misunderstanding on my part.

**Limitations:**

yes.

**Strengths And Weaknesses:**

Note: I use ```this format``` to denote direct quotes from the paper.

Strengths:

- This paper explores the capabilities of LLM agents in various settings, which strengthens its findings and results

Weaknesses:
- While there are few typos ( ```Swe``` and ```(e.g., ,``` are some examples. Moreover, I cannot make sense of the second sentence in the paragraph ```Scenario generation and validation``` due to formatting/typos), my main criticism is that there are some choices of notation that are either never used or overwritten later. For example, $\mathcal{A}$ is used to denote the agent in Section 3.2, however this notation is never used again in the paper. In Section 3.3, it seems like $\mathcal{A}$ is dropped in favor of $\Pi$ for the agent, however $\Pi$ along with outputs $\hat{i}$ and $\mathcal{E}$ for the agent, which are introduced in Section 3.3, are also never used again. To give other examples:
  - In the beginning of Section 5, goals $\textbf{G}1, \textbf{G}2, \textbf{G}3$ are introduced and never used again.
  - In Section 3.3, notation $(A,B,C)$ is introduce to help reason about the steps the agent takes to perform the leakage attack. When I first read this paper, I thought this notation would be used throughout the experimental section, especially in Section 5, where the paper talks about using agents in real world settings. This ABC notation is not used anywhere expect in 2 paragraphs in Section 3.3
- This paper could also benefit from further explanations and further elaborations in many areas. See the Questions section where I go into detail.

---

> ### Author Rebuttal · Authors · 2026-03-31
>
> We thank the reviewer for your careful reading, constructive feedback. We have provided our point-by-point responses below, and we hope they address the concerns.
>
> **Benchmark construction(Q1, Q2)**
>
> In the synthetic benchmark, each pair shares exactly one person. In Netflix, the setup is different: for each user, we create one noisy auxiliary trace and ask which of 1,000 anonymized users it matches. Thus, each Netflix task has one correct answer, but it is not organized as a paired-table instance with exactly one shared person.
>
> For the synthetic benchmark, we generate 60 underlying dataset pairs (20 per fingerprint type). The intent and knowledge conditions reuse these same data and vary only the interaction script, yielding 180 total cases.
>
> **Table 1 (Q3.Q4)**
>
> T  denotes the date tolerance in the classical baseline: an auxiliary rating matches an anonymized record only if the dates are within ±T days. We report mean ± std across three runs below and note one correction to Table 1: the Claude 4.5 value at m=4 should be 97.27, not 64.5, due to a parsing error in one run.
>
> (Please refer to the table Mean ± Std Table in Reviewer2 section due to word limits)
>
> Variance is largest at ((m=2)), where many candidate matches remain plausible. Paired t-tests between GPT-5 and the classical baseline ((T=14)) across the three runs for each (m). The difference is significant at (m=2) ((p=0.034)) and (m=4) ((p=0.008)), but not at (m=6) or (m=8), where both methods reach ceiling performance.
>
> **Cross-model difference(Q5)**
>
> In Table 1, especially at (m=2), the gap appears to reflect disambiguation strategies: GPT-5 continues disambiguation using weaker cues, whereas Claude 4.5 leaves such cases unresolved. In Table 2, the IMPLICIT gap appears to have a different cause: without an explicit linkage request, GPT-5 stays within the benign framing, while Claude 4.5 more often cross-references the datasets and conduct re-identification.
>
> **Utility definition(Q6)**
>
> The utility task in each benchmark instance is a scenario-specific workflow; for example, an agent analyzes employee-training data, write a report, and upload it to OwnCloud. Our utility metric measures whether the agent completes this workflow, following prior agent-evaluation practice \[1\]. We do not separately evaluate report quality, since that requires ground-truth answers or human judgment. Our evaluation captures the utility failure and provides a lower bound on utility cost.
>
> \[1\]AGENTDAM
>
> **AOL casses clarification(Q7)**
>
> We did not intend to claim that only one AOL user was identifiable. Our point is that the most clearly documented public case is the New York Times identification of user No. 4417749 (Thelma Arnold), and the report states that AOL knew of no other identified cases at the time.
>
> **Anthropic case-study (Q8)**
>
> We appreciate the reviewer for pointing this out. We agree that we should have stated more explicitly that our identified count matches Li (2026), and we will revise accordingly. We cannot determine whether our 6 cases are the same, since Li (2026) does not disclose the specific identities.
> Our contribution on this dataset is not the count itself but the analysis. Li demonstrates that re-identification is possible through a case-driven workflow. Our work applies a systematic deanonymization framework and characterizes the recurring linkage mechanisms — in particular, method-driven narrowing (where combinations of methodological choices become uniquely identifying) and role/access cues (where seniority and institutional context disambiguate otherwise plausible candidates). These patterns generalize across our other evaluation settings, which is why we include the Anthropic Interview dataset as a case study rather than a standalone contribution.
>
>
>
> **ChatGPT analysis(Q9)**
>
> We appreciate the reviewer's careful reading. We will revise this section to clarify the preprocessing and to better scope the claims.
>
> PII removal. We masked direct identifiers (e.g., full names) with placeholders while preserving contextual cues, and used an LLM judge to verify that retained cases contained no residual direct PII.
>
> Quantitative reporting. We are limited in how much quantitative detail we can provide because we do not have ground-truth identities for the 30 conversations. Unlike the benchmark and Netflix settings, there is no verification key linking anonymized sessions to known individuals, so we cannot systematically verify which narrowing attempts succeed versus produce plausible but incorrect hypotheses.
>
>
> The “37,000 to 1” example. This is an illustrative narrowing chain for one inferred de-anonymization case, not a quantitative summary of all 30 conversations. The initial 37,000 is a coarse candidate-pool estimate for that case, and later cues progressively narrow the pool.
>
> Moreover, we thank the reviewer for pointing out presentation issues. We will revise the paper to make the notation consistent throughout.

---

> > ### Author Rebuttal · Reviewer_YSKc · 2026-04-03
> >
> > ## Regarding benchmark construction (Q1, Q2):
> > Thank you for answering my questions. Please add these details and/or make them more explicit in the main body.
> >
> > ## Table 1 (Q3, Q4):
> > Thank you for including uncertainties. The results are much stronger now.
> >
> > I applaud the authors for pointing out the error in Table 1. But I am confused about what authors meant by ```a parsing error in one run``` regarding the initial error in Table 1. This is presumably some sort of bug in the parsing code, which would impact all runs, not just one. It would have been nice if the authors reassured me and the other reviewers that they checked the rest of the results of the paper for parsing errors.
> >
> > ## Cross model difference (Q5):
> > Did authors check the reasoning choices for their claims? Such as claiming “GPT-5 continues disambiguation using weaker cues, whereas Claude 4.5 leaves such cases unresolved”, or are these hypothesis?
> >
> > ## Utility definition (Q6):
> > Thank you for answering my questions.
> >
> > ## Aol (Q7):
> > Authors should add the NYT article as the citation for two claims. Claim 1 is that one identified user No. 4417749 (Thelma Arnold) was known by AOL, and, much more importantly, Claim 2 that AOL knew of no other identified cases at the time.
> >
> > ## Anthropic (Q8):
> > I am not happy with the authors’ response. They mention that "method-driven narrowing" and "role/access cues" as something that distinguishes their work from Li (2026). However, Li (2026) also used LLM agents with web-tool access for reidentification. In particular, Li (2026) described their process as: ```for each of these interviews, I call a thinking model agent to search for candidate publications that match the described project and return a ranked list (web search enabled).``` I see no reason why the agents in Li 2026 could not use "method-driven narrowing" and "role/access cues". In fact, I would be surprised if the agents did not do this!
> >
> > Both Li (2026) and this paper gave the entire anonymized interview as context for the agent, and both Li (2026) and this paper allowed the agent to search the web for further auxiliary data. In my opinion, the **only** difference between Li (2026) and this work is how adversarial the prompt is. Li (2026) explicitly asked the agent to link a published scientific work to the interview. This paper explored the benign setting where the agent is tasked with (quoting from the paper being reviewed) ```(i) summarize each interviewee’s technical profile and research niche and (ii) propose nonidentifying research-area descriptors.```
> >
> > It is frustrating that the authors misunderstood the work of Li (2026),  given how similar this work is to the paper.
> >
> > ## ChatGPT analysis(Q9)
> >
> > I am not happy with the author’s response, though here it appears to stem from a misunderstanding of my question, which I do not fault the authors for. My specific question was: in how many of the 30 conversations was the agent able to narrow down the list from 37,000 potential candidates to one? For the purposes of this question, I do not care about whether the agent correctly identified the candidate: I know the authors do not have access to ground truth. What I was curious about was in how many of the 30 conversations was the agent able to narrow down the candidates to one? And, if the agent was not able to narrow down the candidates to one, on average, how many was it able to narrow it down to?
> >
> > It seems that my comment on Table 3 went unanswered, and I wish to emphasize it again. The author’s comment in the rebuttal that ```The "37,000 to 1” example …  is an illustrative narrowing chain … not a quantitative summary of all 30 conversations``` perfectly illustrates my problem with Table 3: it contains no quantitative summary of the 30 conversations, and it should.
> > One thing authors could do is pick a specific conversation, and add a column to Table 3 showing how much each specific linkage (institutions, research domain, geographical regions) narrows down the candidate pool. For example, the number of potential candidates goes from 37,000 to 1,000 after all but a few institutions are linked to the conversation.
> >
> >
> > ## Summary
> > I remain unconvinced by this paper. The paper was submitted with typos, completely unused mathematical definitions, unclear experimental details, and a lack of proper comparison with the related work of Li (2026). After the rebuttal, authors responded to most of these critiques, including promising to fix/remove their various unused math definitions but gave no details on how, and once again improperly compared to Li (2026).
> >
> > I will keep my score of 2: Reject. This paper needs more polishing before it can be accepted.

---

> > > ### Author Response · Authors · 2026-04-08
> > >
> > > We thank the reviewer for the detailed follow-up and for clarifying the remaining concerns.
> > >
> > > **Q3, Q4: Table 1**
> > > We confirm that we rechecked all reported values. There are no other errors.
> > >
> > > **Q5: Cross-model difference**
> > > We appreciate the reviewer for raising this clarification. This claim is based on manual inspection of the agent trajectories.In the low-information (m=2) setting, we observed a consistent difference in model behavior: GPT-5 more often continues disambiguation using weaker secondary cues, including temporal consistency—i.e., whether the observed rating dates remain plausibly aligned with a candidate user’s rating timeline despite the injected date noise. By contrast, Claude 4.5 appears to rely more heavily on movie overlap and rating consistency as primary signals, and to treat temporal agreement as a weaker supporting cue. As a result, when movie/rating evidence alone does not sufficiently separate candidates, Claude 4.5 more often leaves the case unresolved (e.g., “no match found”), whereas GPT-5 more often proceeds with a best-match hypothesis.
> > >
> > > **Q7: AOL**
> > > We will add the corresponding citation to support the claims regarding the identified AOL user and AOL’s knowledge of other identified cases at the time:  [https://web.archive.org/web/20180312064453/http://www.nytimes.com/2006/08/09/technology/09aol.html](https://shorturl.at/qSYjp)
> > >
> > > **Q8: Anthropic case study and relation to Li (2026)**
> > >
> > > In our case study, the re-identification process can involve broader researcher-level information, whereas Li (2026), as described in the paper, primarily focuses on matching project-related information in the interview to public scientific works. In one representative case, for example, the interviewee’s high-level self-description in the interview as a researcher aligned closely with a public professional profile. This difference may reflect differences in task framing and prompt setup, but because Li (2026) intentionally omits many operational details for privacy reasons, we cannot make a stronger mechanism-level comparison.
> > >
> > > We agree that our Anthropic Interview case study is similar to Li (2026): both works show that web-enabled LLM agents can support re-identification from anonymized interview transcripts, and we will revise the paper to make this overlap explicit.
> > >
> > > That said, this case study is not our main contribution. Our primary contribution is the controlled benchmark and evaluation framework for inference-driven linkage, which enables systematic, reproducible, and scalable evaluation of this risk across fingerprint structure, task framing, and attacker knowledge. We include the Anthropic Interview case study to test whether inference-driven linkage also appears in a realistic unstructured setting, not to make a separate deanonymization capability claim relative to Li (2026).
> > >
> > > **Q9: ChatGPT log case study**
> > > We would like to clarify that the 30 conversations are used together in a single deanonymization attempt, rather than being treated as 30 separate attempts. They come from a shared account and may correspond to at most 11 underlying users; because no ground-truth user mapping is available, we do not know how many conversations belong to each user. To allow the agent to aggregate cues across logs, we provided the full set of conversations jointly and asked whether a web-enabled agent could re-identify the underlying user(s) from the combined evidence. Under this setup, the relevant unit of analysis is the aggregate log collection, not each individual conversation. Accordingly, we cannot report how many of the 30 conversations, taken individually, would narrow the candidate pool to one, because that was not the evaluation unit in our experiment.
> > >
> > > We agree, however, that Table 3 would be stronger with a more explicit quantitative narrowing example. We will revise it to make clear that the table illustrates one representative narrowing chain over the combined logs, and we will add approximate candidate-pool sizes at each stage:
> > > | Signal added | Approximate candidate pool |
> > > |---|---|
> > > | Institutional context / geographic cue | ~37,000 |
> > > | Role / seniority / domain / department | 200–300 |
> > > | Research specialization | 5–10 |
> > > | Time-aligned activity / internship | 1–2 |
> > > | Publication record | 1 |
> > >
> > > **Notation and presentation.**
> > > The formal definitions in §3.1–§3.3 were introduced to make the deanonymization framework precise and to unify the benchmark and case-study settings under a common problem formulation. Their role is primarily to clarify the framework, rather than to introduce notation that must be reused throughout every experimental section. We agree, however, that the current presentation would benefit from clearer notation use, correction of typos, and elaboration of some experimental details. We will revise the paper to streamline the presentation and improve readability, including simplifying or removing notation that is not essential.

---

### Official Review · Reviewer_QXZ1 · 2026-03-13

**Soundness:** 3
**Presentation:** 4
**Significance:** 3
**Originality:** 2
**Overall Recommendation:** 4
**Confidence:** 4

**Summary:**

This paper evaluates an underconsidered privacy risk for LLM agents, which goes beyond simple “leakage” or data memorization, which is called "inference-driven linkage". The central contribution is to show that the agent can gather several weak clues that, taken individually, are not identifying, to formulate a hypothesis about a person's identity.  The article experimentally illutrates this issue using past attacks (AOL and Netflix) that they replay in the form of a controlled simulation in the considered context, showing results that go beyond those of the classic baseline (the algorithm used at the time). Similarly, they examine a case study on the Anthropic dataset which works in the context of inference-driven linkage, via an agent that can infer fingerprints from anonymized (redacted) interviews and cross-references with the web. The article also proposes a mitigation technique based on a privacy-preserving prompt that instructs the agent handling data potentially subject to inference-driven linkage to refuse de-anonymization requests. They show that this defense greatly reduces leakage, sometimes to almost zero for certain models, but at a cost in terms of utility.

**Compliance With Llm Reviewing Policy:**

Affirmed.

**Final Justification:**

The rebuttal clarifies the empirical contribution and benchmark design, but it only partly resolved my concerns about the robustness of the evaluation metrics. So I maintain a weak accept.

**Key Questions For Authors:**

1- Could you clarify what is new relative to the classical data linkage and deanonymization literature, beyond the agentic setting itself ?

2- How realistic is the proposed benchmark, and to what extent do the results hold in less artificial settings ?

3- To what extent evaluation metrics for privacy and utility are robust or could be generalized, in particular the use of "LLM as a judge" for privacy and "task completion" for utility ?

**Limitations:**

The limitation include the realism of the benchmark, the somewhat ad hoc evaluation metrics, and the lack of mitigation beyond prompt-based ones.

**Strengths And Weaknesses:**

Strength:

- Clear and well-written empirical evaluation paper, with a well-formulated problem, a controlled benchmark and comparisons with serveral case studies.

- The paper highlights a risk in the evaluation of agent privacy which is under-considered, where the agent reconstructs an identity by aggregating personal data by accessing several sources and crossing them based on fingerprints formed with inferable information to establish correspondences. They call this "inference-driven linkage". The formulation as a specific dimension of LLM agent evaluation is relevant and well-motivated.

- A simple benchmark is provided as a methodological contribution. It allows for variation in the fingerprint, the framing of the task and the background knowledge of the attaquer. The paper shows that leakage may occur even without explicit demand for re-identification.

- Several case studies (Netflix, AOL, Entropic interviews, LogChatGPT) illustrate the phenomenon

- A mitigation solution based on simple prompt gives good results, with a privacy-utility tradeoff.

Weakness:

- There is no real technical contribution in terms of new algorithm or learning method, nor any formal guarantee or general mechanism to prevent linkage.

- The conceptual novelty is rather limited or incremental. The fact that aggregating many weakly identifying data can enable a person to be re-identified is not surprising. The novelty mainly lies in proposing a way to measure it.

- The mitigation part is not very ambitious technically, nor is it general. The only defense that is actually evaluated is a simple privacy-oriented system prompt.

- The privacy metrics considered (LLM as a judge) and utility metrics (uploading a file on owncloud) make sense but are ad hoc.

- The benchmark is very synthetic which limits its realism.

- On Netflix, the comparison shows an improvement over old baselines, but not necessarily a convincing improvement over current non-LLM tecniques.

---

> ### Author Rebuttal · Authors · 2026-03-31
>
> We thank the reviewer for your careful reading, constructive feedback. We have provided our point-by-point responses below, and we hope they address the reviewer’s concerns.
>
> **Technical contribution / lack of new algorithm or formal guarantee**
> We agree that our paper is not an algorithm-design paper, and we will clarify this more explicitly. Instead, our contribution is empirical measurement and benchmarking: we identify and quantify a previously underexplored privacy failure mode, inference-driven linkage in LLM agents. We believe this is a meaningful technical contribution, well aligned with ICML, where benchmark and evaluation papers often shape how the community understands emerging risks. Specifically, we contribute: (i) a parameterized benchmark generation pipeline varying fingerprint type, task intent, and attacker knowledge, with ecological-validity validation; (ii) a composable deanonymization evaluation framework spanning structured and unstructured settings; and (iii) the first systematic evidence that linkage can arise as a byproduct of benign agent behavior. Our results also suggest why formal guarantees are difficult and show a privacy–utility trade-off: because linkage arises from general reasoning over scattered cues rather than memorization, mitigation can reduce leakage but substantially degrade task performance.
>
> **Conceptual novelty relative to classical linkage/deanonymization**
>  We agree that the general principle—aggregating weak identifiers can enable re-identification—is well established; our novelty lies in three findings. First, agents can generate identity hypotheses as a byproduct of benign tasks, unlike classical linkage attacks that assume explicit adversarial intent. Second, they remove the engineering barrier: on Netflix, a generic prompt reaches 79.2% vs. 56.0% for the classical heuristic with 2 fragments, without domain-specific tuning. Third, our benchmark is the first reproducible framework to quantify this risk systematically rather than leaving it anecdotal.
>
> **Mitigation is simple and not general**
>  The prompt defense is simple by design: its role is diagnostic, not to claim a general solution. It shows two things that would otherwise be missing: linkage can be suppressed by changing agent behavior, and doing so incurs measurable utility costs (e.g., Claude 4.5 drops from 0.989 to 0.455 in Explicit-MK), making the privacy–utility trade-off explicit. More ambitious mitigations such as runtime monitoring, tool gating, or training-time interventions are important next steps, and we will discuss them more explicitly.
>
> **Privacy and utility metrics may seem ad hoc**
> These metrics are not ad hoc; they are matched to the tool-using agent setting and consistent with prior agent-evaluation practice [1]. For privacy, we use an LLM judge over the full trajectory with two criteria—explicit identification and attributional linkage—because simpler alternatives such as string matching miss attributional linkage. For utility, end-to-end task completion is the relevant measure because the benchmark asks whether privacy-aware defenses break the full workflow. Richer privacy and utility metrics are natural extensions, which we will discuss.
>
> [1] AgentDAM: Privacy Leakage Evaluation for Autonomous Web Agents
>
> **The benchmark is synthetic and may lack realism**
>  The benchmark is synthetic because its purpose is causal isolation, not ecological simulation: real-world settings confound noise, candidate pool size, format, and task framing. Synthetic construction lets us hold these fixed while varying fingerprint type, task intent, and attacker knowledge, and each instance is validated with a 9-point checklist(Appendix D). Our conclusions do not rest on the benchmark alone: Netflix, AOL, and the case studies provide real-data or real-trace evidence, while the benchmark explains when and why linkage occurs under controlled conditions.
>
>  **Netflix compares against older baselines, not necessarily current non-LLM techniques**
> The Netflix experiment is not meant to benchmark against all current non-LLM linkage methods, but to show that the engineering barrier has changed: the classical approach relied on hand-crafted scoring and tuned tolerance parameters, whereas the agent achieves comparable or stronger performance with only a generic prompt and no heuristics, feature engineering, or tuning. Our claim is therefore not state-of-the-art accuracy, but strong linkage at near-zero engineering cost.
>
>  **Realism beyond the benchmark / AOL selection**
> The AOL experiment measures agent capability, not a population-level re-identification rate. We first remove histories with self-PII, then filter for cases where inference-driven linkage is plausibly testable; most histories are simply too sparse to support linkage. Even under this favorable selection, the task remains non-trivial—agents fail on 75% of cases—yet still recover substantially more identifications than prior manual analysis.

---

> > ### Author Rebuttal · Reviewer_QXZ1 · 2026-04-04
> >
> > The rebuttal addresses some of my concerns in a helpful way the intended contribution as an empirical benchmarking and evaluation study rather than an algorithmic one, explains that the bechmark is kept synthetic as a tool for causal isolation rather than realism, better justifies the Netflix comparison and the prompt-based mitigation. On the other side, I remain only partially convinced on the robustness and generalizability of the evaluation metrics, especially the reliance on an LLM judge for privacy assessment and task completion as the main utility measure, with limited evidence of their reliability. Overall, the rebuttal improves the paper but I maintain my current weak accept assessment, because it does not fully resolve the methodological concerns.

---

> > > ### Author Response · Authors · 2026-04-08
> > >
> > > We thank the reviewer for this constructive follow-up. To support the reliability of our metrics, we provide additional evidence for the privacy judge and clarify the scope of the task-completion utility measure.
> > >
> > > **Privacy metric.** Because the full benchmark contains 27 settings (3 models × 3 fingerprint types × 3 task framings: implicit, explicit-ZK, and explicit-MK), with 20 cases per setting (540 runs total), exhaustive human validation is costly. We therefore conducted two reliability checks on one representative setting (GPT-5 / intrinsic / implicit). First, we performed a human-validation study on 20 examples and measured agreement using Cohen’s κ: the two human annotators achieved high agreement (κ = 0.83), and the LLM judge showed substantial agreement with the human annotators on average (κ = 0.72). This indicates that the judge is reasonably reliable for detecting privacy violations in this setting. Second, we reran the same setting three times to assess stability: the leakage rate was 13.33% ± 2.89%, compared to 15% in the original run. Together, these results suggest that the privacy metric is reasonably stable and reliable for our comparative evaluation.
> > >
> > > **Utility metric.** Our goal is to measure whether privacy-preserving interventions disrupt the agent’s ability to complete the intended end-to-end workflow. Binary task completion directly captures this form of utility loss. Evaluating report content or quality would address a different question and would require an additional task-specific evaluator (e.g., human judgment or a separate LLM judge), reducing comparability across benchmark conditions. We therefore use task completion as the utility measure because it is directly aligned with our evaluation goal and consistent with prior agent-evaluation work such as AgentDAM, which measures utility via task success in the environment. We will clarify this scope more explicitly in the revision.

---

### Official Review · Reviewer_YnNX · 2026-03-13

**Soundness:** 3
**Presentation:** 3
**Significance:** 4
**Originality:** 2
**Overall Recommendation:** 4
**Confidence:** 4

**Summary:**

This paper studies privacy risks that arise when LLM agents perform multi-step reasoning over fragmented information. The authors argue that agentic reasoning enables automated data linkage attacks, where individually non-identifying signals can be aggregated to reconstruct identities. The paper evaluates this claim in three settings. First, the authors reproduce classical de-anonymization attacks (e.g., Netflix and AOL datasets) and show that LLM agents can automate much of the reasoning previously performed manually. In their experiments, agents recover identities for a large fraction of users in the Netflix dataset, outperforming heuristic linkage methods. Second, the paper introduces a controlled benchmark where identity traces are represented by partial attribute fragments. Agents are tasked with synthesizing hypotheses about potential identities based on these clues. Finally, the authors examine unstructured real-world traces to evaluate whether agents can perform similar inference under noisy conditions.

Overall, the paper argues that agentic reasoning significantly lowers the barrier to inference-driven privacy attacks, even when individual data points appear benign.

**Compliance With Llm Reviewing Policy:**

Affirmed.

**Key Questions For Authors:**

1. Could the authors clarify the agent architecture used in the experiments? In particular, is the agent allowed to perform tool use, retrieval, or iterative reasoning steps beyond a single prompt?

2. Beyond privacy-aware prompts, have the authors considered system-level mitigations such as monitoring reasoning trajectories for identity inference patterns?

**Limitations:**

yes

**Strengths And Weaknesses:**

Strengths
1. The paper highlights an important emerging privacy risk. Most existing LLM privacy work focuses on training data memorization, prompt extraction and direct leakage of training data. This paper instead focuses on inference-driven privacy leakage, where identities are reconstructed by aggregating weak signals. This is a meaningful extension of existing privacy threat models to modern AI systems.
2. The experiments showing that agents can reproduce classical linkage attacks are interesting. The Netflix experiment, in particular, provides a concrete illustration that reasoning agents can automate what previously required human analysis. The controlled benchmark for fragmented identity traces is also a useful attempt to standardize evaluation of inference-based privacy risks.

Weaknesses
1. This is basically applying LLM agents to an already known data linkage attack. The paper does not introduce fundamentally new attack mechanisms. Instead it shows that agents make existing attacks easier. That is interesting but not groundbreaking.
2. Agent architecture and system setup are not fully clear. Since the central claim concerns agentic reasoning, the paper would benefit from clearer details about the system configuration. In particular: What agent framework is used? Does the agent use tools or external retrieval? How many reasoning steps are allowed? These details matter because they directly affect the power of the attack.

---

> ### Author Rebuttal · Authors · 2026-03-31
>
> We thank the reviewer for your careful reading, constructive feedback. We have provided our point-by-point responses below, and we hope they address the reviewer’s concerns.
>
> **Novelty and contribution.**
>
> We appreciate the reviewer's engagement and would like to clarify what we see as new beyond making existing attacks easier.
>
> The core novelty is not that agents can replicate classical linkage — it is that agents introduce a failure mode that classical attacks do not capture: **inference-driven linkage as an emergent byproduct of routine tasks.** In Table 2, under implicit (benign) intent, Claude 4.5 produces identity-level leakage in 77% of cases without ever being asked to identify anyone. This is not a known attack being automated — it is a new category of privacy risk where linkage arises from an agent being helpful, not from an adversary being clever. Classical data linkage research assumes an attacker with intent and expertise; our results show that neither is required.
>
> Second, the artifact types where this risk arises are themselves new. Classical linkage attacks targeted structured databases (rating tables, medical records). We show that agents can perform linkage over unstructured traces — interview transcripts, chat logs, search histories — by synthesizing scattered contextual cues that do not map to any predefined schema. This is qualitatively different from table-to-table matching and was not feasible without the inferential capabilities that modern agents provide.
>
> Third, we contribute the first controlled benchmark for this risk, enabling systematic measurement across fingerprint types, intent framings, and attacker knowledge levels. This moves the field beyond case-specific demonstrations toward reproducible, comparable evaluation — something that did not exist for inference-driven linkage prior to our work.
>
> We believe these contributions go beyond showing that agents make existing attacks easier: they identify a new threat vector (byproduct linkage), demonstrate it on new artifact types (unstructured traces), and provide the first systematic framework for measuring it.
>
> **Agent architecture**
>
> Yes. In all settings, the agent operates over multiple turns and can perform iterative reasoning — forming hypotheses, executing actions, and refining based on results. For Netflix and the controlled benchmark, we use OpenHands together with The Agent Company, where the agent has access to code execution, file handling, and report generation/uploading, and the auxiliary evidence is fixed and provided upfront. For AOL, the Anthropic Interview case study, and the ChatGPT log case study, we use a web-enabled Gemini interface, since these settings require the agent to retrieve and corroborate auxiliary evidence dynamically via web search. The architecture differs across settings because the structure of the auxiliary evidence differs — fixed tables versus open-web retrieval — but in both cases the agent reasons iteratively beyond a single prompt. We do not fix the number of reasoning steps as a separate parameter; the agent is allowed to continue as needed within the execution environment, including retrying after failed actions, until it either completes the task or terminates. We will clarify this more explicitly in the revision.
>
> **System-level mitigation.**
>
> We thank the reviewer for this suggestion. We focused on privacy-aware system prompts as a simple, deployable first-line defense that requires no infrastructure beyond the agent itself. System-level monitoring of reasoning trajectories — for example, detecting when intermediate steps begin to converge on an identity hypothesis — is a promising complementary direction that could enforce privacy constraints at runtime without relying on the model's own compliance. We will discuss this and other system-level mitigations (e.g., output filtering, tool-access gating based on data sensitivity) as future work in the revision.
>
> ***Mean ± Std for Netflix Experiments***
> | M | Claude 4.5 | GPT-5 |
> |---|---:|---:|
> | 2 | 53.30 ± 19.21 | 79.17 ± 4.97 |
> | 4 | 97.27 ± 2.78 | 94.83 ± 2.22 |
> | 6 | 93.13 ± 8.39 | 97.43 ± 3.76 |
> | 8 | 97.30 ± 3.50 | 99.00 ± 0.72 |

---

> > ### Author Rebuttal · Reviewer_YnNX · 2026-04-03
> >
> > Thanks for the detailed response. Regarding ``We show that agents can perform linkage over unstructured traces — interview transcripts, chat logs, search histories'': I thought this has already been demonstrated in a paper by ETH group a year ago (though I forget the title of that paper) and another similar paper: https://arxiv.org/pdf/2602.16800 (I know this one is post ICML submission deadline).

---

> > > ### Author Response · Authors · 2026-04-08
> > >
> > > We thank the reviewer for this follow-up. We will acknowledge Large-scale online deanonymization with LLMs as concurrent work in the revision.
> > >
> > > Our main contribution is not another case-driven demonstration of deanonymization over unstructured traces, but a controlled and scalable evaluation framework for studying inference-driven linkage. Specifically, we introduce a benchmark that systematically varies fingerprint structure, task intent, and attacker knowledge, enabling reproducible measurement of when identity linkage emerges—including under benign (implicit) task settings.
> > >
> > > The unstructured case studies (AOL, interview transcripts) are included to show that this risk manifests in realistic settings.
> > > We agree that prior work, including ETH work on inference from text and concurrent efforts such as arXiv:2602.16800, shows that LLMs can infer or recover identity-related information from unstructured data. Our work is complementary: rather than adding another instance, we provide a framework to systematically measure and compare this risk across conditions, including settings where linkage arises without explicit re-identification intent.
> > >
> > > If the reviewer had a specific paper in mind, we would appreciate the reference and will include a targeted comparison.

---

### Official Review · Reviewer_NZHG · 2026-03-13

**Soundness:** 3
**Presentation:** 3
**Significance:** 3
**Originality:** 3
**Overall Recommendation:** 4
**Confidence:** 3

**Summary:**

This paper studies the privacy risk of inference-driven linkage, where LLM-based agents reconstruct identities by aggregating fragmented signals that are individually non-identifying. The authors propose a unified evaluation framework and test it in three settings: reproducing the classical Netflix and AOL de-anonymization cases, introducing a controlled benchmark, and analyzing real-world unstructured traces. The paper also evaluates a privacy-aware system prompt and shows a privacy–utility trade-off.

**Compliance With Llm Reviewing Policy:**

Affirmed.

**Final Justification:**

The rebuttal has addressed my main concerns in a meaningful way. In particular, it clarifies the intended role of the controlled benchmark versus the end-to-end evaluations, explains that the AOL filtering is meant to measure agent capability rather than population-level risk, and provides a more concrete analysis of cross-model differences through refusal rates. The response improves the clarity and credibility of the paper. I therefore appreciate the revision and would be willing to increase my score.

**Key Questions For Authors:**

Q1. To what extent does the current benchmark reflect real-world agent settings, where auxiliary information is noisier and multiple plausible matches often exist?

Q2. In the AOL search log experiment, the benchmark further selects 40 high-salience histories using an LLM judge. It would be helpful to clarify whether this filtering step is too strong, since selecting cases that already contain strong identifying cues may inflate the measured linkage success rate compared with more typical query histories.

Q3. The authors should provide a deeper analysis of why different agents exhibit such large differences in linkage behavior under the same conditions. For example, in Table 2, could the authors explain more clearly why COORDINATE signals are highly linkable in Explicit-ZK but become less risky than INTRINSIC and HYBRID for GPT-5 in Explicit-MK?

**Limitations:**

The paper discusses some limitations implicitly, such as the abstraction gap between controlled benchmarks and real-world traces, and the privacy-utility trade-off of the prompt-based defense, but it does not provide a clear and consolidated limitations section.

**Strengths And Weaknesses:**

Strengths
1. The paper addresses an important privacy problem by extending the discussion from direct leakage to reasoning driven de-anonymization, which is relevant for modern retrieval augmented agents.
2. The evaluation framework is fairly unified and covers both classical linkage settings as well as more open ended and realistic scenarios.
3. The benchmark is well structured, and the paper considers both privacy leakage and utility, which makes the discussion of the trade-off more meaningful.
Weakness
1. While the paper raises an important and timely privacy concern, some parts of the benchmark remain somewhat stylized.
2.The evaluation relies heavily on an LLM judge for identifying leakage events, but the paper does not provide enough validation against human annotation to fully establish the reliability of this measurement.
3. Although the paper demonstrates that benign task framing can still trigger linkage behavior, it would benefit from a deeper analysis of why different agents behave so differently across the same conditions.

---

> ### Author Rebuttal · Authors · 2026-03-31
>
> We thank the reviewer for the constructive feedback. We have provided our point-by-point responses below, and we hope they address the reviewer’s concerns.
>
> **Benchmark realism**
>
> Both concerns are captured by the end-to-end evaluations rather than isolated in the controlled benchmark. In our Netflix experiments (§4.1.1, Table 1), agents identify the correct user from a pool of 1,000 with many plausible alternatives, using auxiliary traces perturbed by ±14 days and ±1 star with 50% probability. Even with only 2 rating fragments, agents achieve 79.2% linkage success versus 56.0% for the classical heuristic. The AOL and case studies (§4.1.2, §5) present an even harder setting: agents operate over unstructured traces, corroborating ambiguous cues through open-web retrieval against an unbounded candidate population.
>
> The benchmark serves a complementary purpose. In the settings above, outcomes could reflect noise levels, candidate pool structure, fingerprint type, or task framing — these factors vary simultaneously. The benchmark resolves this by holding noise and candidate structure fixed, so that differences in linkage rates are attributable to the variable being manipulated. The single ground-truth overlap among 10 records ensures unambiguous linkage rather than incidental duplicates.
>
> In the revision, we will make these complementary roles more explicit.
>
> **AOL filtering** We would like to clarify that the AOL experiment is designed to measure agent capability, not a population-level re-identification rate, and we will make this framing more explicit in the revision. The pipeline first removes histories containing self-PII (e.g., resumes, full names), so the retained cases rely on inference-driven linkage rather than trivial identifier leakage. The subsequent salience filtering selects histories where this capability is meaningfully testable, since most query histories are too generic to support linkage (e.g. broad local-interest, or school/community queries only). The resulting 10/40 success rate should therefore be read as: when individually non-identifying but combinable cues are present, agents can exploit them to reconstruct identity. We note that even under this favorable selection, the task remains non-trivial — agents fail on 75% of the filtered cases — yet still recover 10× more identifications than prior manual analysis.
>
> **Why models differ, and why COORDINATE drops for GPT-5 in EXPLICIT-MK.**
>
> We thank the reviewer for this question. We provide new evidence — refusal rates — that helps explain the observed patterns, and focus first on the GPT-5 anomaly before addressing broader cross-model differences.
>
> **The GPT-5 COORDINATE anomaly.** The key explanatory variable is refusal behavior. In our trajectory analysis, GPT-5's refusal rate on COORDINATE cases jumps from .00 in Explicit-ZK to .25 in Explicit-MK, while INTRINSIC and HYBRID remain at .05. This accounts for most of the drop in linkage success (.900 → .650). We hypothesize that coordinate signals, such as locations and timestamps, combine with a named target in Explicit-MK to resemble surveillance or tracking more closely, which safety training likely flags more aggressively. In contrast, intrinsic signals (behavioral traits, preferences) feel more like routine data analysis even when a name is attached, and thus trigger less refusal.
>
> **Cross-model variation.** Refusal rates also explain the broader model differences. Claude 4.5 shows near-zero refusal across conditions, consistent with uniformly high linkage success. GPT-5 exhibits selective refusal that is sensitive to the interaction between signal type and intent framing. o4-mini shows similarly low refusal, but lower linkage success, suggesting weaker matching capability — missing the correct shared cues or choosing the wrong candidate — rather than safety-driven restraint.
>
> We include the full refusal rates below and will add this analysis in the revision.
>
> | Intent | Fingerprint | GPT-5 (LSR) | GPT-5 (Ref) | o4-mini (LSR) | o4-mini (Ref) | Claude 4.5 (LSR) | Claude 4.5 (Ref) |
> | :--- | :--- | ---: | ---: | ---: | ---: | ---: | ---: |
> | EXPLICIT-ZK | Intrinsic | .800 | .00 | .650 | .00 | .900 | .10 |
> | | Coordinate | .900 | .00 | .400 | .00 | .850 | .05 |
> | | Hybrid | .850 | .10 | .500 | .00 | 1.000 | .05 |
> | EXPLICIT-MK | Intrinsic | .950 | .05 | .750 | .00 | 1.000 | .00 |
> | | Coordinate | .650 | .25 | .600 | .20 | .900 | .00 |
> | | Hybrid | .950 | .05 | .800 | .10 | 1.000 | .00 |
>
> **Limitations**
>
> We thank the reviewer for this suggestion. We agree that a consolidated limitations section would improve clarity. In the revision, we will add an explicit limitations section that brings together the points currently discussed across the paper.

---

> > ### Author Rebuttal · Reviewer_NZHG · 2026-04-02
> >
> > The rebuttal has addressed my main concerns in a meaningful way. In particular, it clarifies the intended role of the controlled benchmark versus the end-to-end evaluations, explains that the AOL filtering is meant to measure agent capability rather than population-level risk, and provides a more concrete analysis of cross-model differences through refusal rates. The response improves the clarity and credibility of the paper. I therefore appreciate the revision and would be willing to increase my score.

---

> > > ### Author Response · Authors · 2026-04-03
> > >
> > > Dear Reviewer NZHG,
> > >
> > > We are happy to hear that our responses have addressed your concerns and questions. We appreciate you taking the time to read our rebuttal and adjust your evaluation accordingly. We will incorporate all the clarifications and additional analysis discussed during the rebuttal into our revised version. Thank you once again for your valuable, constructive feedback and for your consideration.
> > >
> > > Best regards,
> > > Authors

---

### Decision · Program_Chairs · 2026-04-30

**Decision:**

Accept (regular)

**Comment:**

This paper studies inference-driven linkage attacks in which modern agents aggregate weak, individually non-identifying signals to reconstruct identities. Reviewers generally agreed that this is an important emerging privacy risk. The paper was viewed as well motivated, and reviewers appreciated the combination of classical case studies, a controlled benchmark, and a mitigation analysis that explicitly considers the privacy-utility tradeoff.

The main concerns were about framing and evaluation details rather than the central phenomenon itself. Reviewers wanted stronger discussion of what is new relative to classical linkage work, better validation of the LLM-based evaluation components, and clearer acknowledgement of benchmark limitations and realism gaps. I read the discussion and rebuttal, and it appears that the authors addressed a substantial portion of these concerns, especially around the intended role of the controlled benchmark, the interpretation of the AOL setting, and the reliability of the privacy metric. Given the overall positive reception and the importance of the problem, I recommend acceptance.